# The Addition of High Doses of Hyaluronic Acid to a Biphasic Bone Substitute Decreases the Proinflammatory Tissue Response

**DOI:** 10.3390/ijms20081969

**Published:** 2019-04-22

**Authors:** Dominik Sieger, Tadas Korzinskas, Ole Jung, Sanja Stojanovic, Sabine Wenisch, Ralf Smeets, Martin Gosau, Reinhard Schnettler, Stevo Najman, Mike Barbeck

**Affiliations:** 1Department of Oral and Maxillofacial Surgery, Division for Regenerative Orofacial Medicine, University Hospital Hamburg-Eppendorf, 20246 Hamburg, Germany; siegerdominik@gmail.com (D.S.); tadaskorzinskas@yahoo.de (T.K.); ol.jung@uke.de (O.J.); r.smeets@uke.de (R.S.); m.gosau@uke.de (M.G.); reiner.schnettler@mac.com (R.S.); 2Department for Cell and Tissue Engineering, Institute of Biology and Human Genetics, University of Niš, Faculty of Medicine, Niš 18106, Serbia; sanja.stojanovic@medfak.ni.ac.rs (S.S.); stevo.najman@medfak.ni.ac.rs (S.N.); 3Clinic of Small Animals, c/o Institute of Veterinary Anatomy, Histology and Embryology, Justus Liebig University of Giessen, 35392 Giessen, Germany; Sabine.Wenisch@vetmed.uni-giessen.de; 4Department of Oral Maxillofacial Surgery, University Medical Center Hamburg-Eppendorf, 20246 Hamburg, Germany; 5BerlinAnalytix GmbH, 12109 Berlin, Germany

**Keywords:** hyaluronic acid, biphasic bone substitute, biocompatibility, tissue reaction, inflammation, macrophage, M1, M2, multinucleated giant cells

## Abstract

Biphasic bone substitutes (BBS) are currently well-established biomaterials. Through their constant development, even natural components like hyaluronic acid (HY) have been added to improve both their handling and also their regenerative properties. However, little knowledge exists regarding the consequences of the addition of HY to their biocompatibility and the inflammatory tissue reactions. Thus, the present study was conducted, aiming to analyze the influence of two different amounts of high molecular weight HY (HMWHY), combined with a BBS, on in vitro biocompatibility and in vivo tissue reaction. Established in vitro procedures, using L929 cells, were used for cytocompatibility analyses under the test conditions of DIN EN:ISO 10993-5. For the in vivo part of the study, calvarial defects were created in 20 Wistar rats and subsequently filled with BBS, and BBS combined with two different HMWHY amounts, i.e., BBS + HY(L) and BBS + HY(H). As controls, empty defects were used. Established histological, immunohistochemical, and histomorphometrical methods were applied to analyze the tissue reactions to the three different materials, including the induction of pro- and anti-inflammatory macrophages and multinucleated giant cells (BMGCs). The in vitro results showed that none of the materials or compositions caused biological damage to the L929 cells and can be considered to be non-toxic. The in vivo results showed that only the addition of high doses of HY to a biphasic bone substitute significantly decreases the occurrence of pro-inflammatory macrophages (* *p* < 0.05), comparable to the numbers found in the control group, while no significant differences within the three study groups for M2-macrophages nor BMGCs were detected. In conclusion, the addition of different amounts of HMWHY does not seem to affect the inflammation response to BBS, while improving the material handling properties.

## 1. Introduction

Bone substitute materials (BSM) are in daily use for different applications in the medical fields of orthopedics, traumatology, and dentistry. Most often, BSM are used in form of granules or blocks, while for other applications, such as sinus augmentations, extraction sockets, bone cysts, or implantation sites around a screw for augmentation, bone substitute pastes (BSP) are more suitable, due to their improved handling properties [1]. In addition to the “classic” BSM in granular form, different hydrophilic and water-binding molecules have already been included into BSP to achieve flowability and, thus, to improve their applicability [2]. The bone substitute granules that are part of a BSP are most often based on calcium phosphates that are of either natural (NBSM) or synthetic origin (SBSM) [3]. SBSM in particular are used as basis for BSP, as these materials can easily be produced with suitable properties, such as the required granule size for minimally invasive applications via a syringe [4].

In this context, it has been shown that the application of SBSM based on pure hydroxyapatite (HA)- or pure beta-tricalcium phosphate (β-TCP)-based granules has been associated with disadvantages regarding the biodegradation processes [5]. Thus, HA and β-TCP have different solubility behaviors, and induce different levels of a material-induced foreign body response, which includes phagocyting cells, such as macrophages or biomaterial-induced multinucleated giant cells (BMGCs) [6]. While HA-based materials have shown to exhibit low dissolution and biodegradation behaviors, BSM based on β-TCP becomes degraded very quickly via both processes [7]. To overcome these opposite biodegradation behaviors, both compounds have been mixed in different ratios to become what is called biphasic bone substitutes (BBSM), and it has already been revealed that the mixture of HA and β-TCP combines their respective biological tissue responses [8]. In this context, it has been determined that the mixture of both compounds with a HA/β-TCP ratio of 60/40 wt% induced a tissue reaction including a high BMGC formation and vascularization rate comparable to pure β-TCP within the initial time span after implantation, while later the tissue reaction was comparable to the HA-group [5,9]. Altogether, these and other different preclinical and clinical results substantiate the applicability of BBSM, as their biodegradation rate has shown to correlate with the process of bone tissue regeneration [10,11]. Moreover, their degradation pattern has been demonstrated to comply with the process of creeping substitution [5,12,13]. Thus, BBSM are also assumed to be a suitable component of a BSP.

The next component of a BSP is a liquid, or one with more hydrophilic molecules, that show rheologic properties after the addition of blood or water [2]. Different natural-based polymers, such as cellulose and its derivatives, collagen from different sources, or hyaluronic acid (HY), amongst a variety of other polymers, have already been used as component of different BSPs [14,15,16,17,18,19,20]. Beside improving the rheologic properties of a BSP, these molecules have been suspected to provide additional regenerative features [4]. Extracellular matrix proteins in particular, such as collagen or HY, are considered key players in the tissue regeneration process by providing a matrix for (bone) tissue growth or ingrowth of vessels, and moreover by interacting with different cell types, such as osteoblasts [21,22,23]. Even HY, as a member of the glycosaminoglycan (GAG) family, has been proven to modulate wound healing and related processes, such as inflammation, cellular migration, and angiogenesis via specific receptors [24,25,26,27,28,29]. In the context of bone regeneration, it has been revealed that HY positively influences the vascularization within the implantation bed of a BSP, and also osteoblastic growth [30]. Interestingly, it has also been shown in a preclinical in vivo study, using the subcutaneous implantation model in Wistar rats that an injectable paste-like material containing β-TCP granules and HY served as a stable and barrier-like structure over a period of at least 60 days, without allowing premature cell and tissue invasion inside the central implant regions [4]. Thus, material degradation was shown to take place from the peripheral area towards the center of the implantation bed over time. Based on these results, it was assumed that the BSP featured an integration behavior that is in accordance with the concept of the guided bone regeneration (GBR). Furthermore, the performance of the same biomaterial was analyzed for bone tissue regeneration in a preclinical and clinical study [30,31]. The results of these two studies showed that the BSP contributes to sufficient bone regeneration by serving as a scaffold-like structure. 

However, in these studies, the influence of the addition of HY on the material-associated inflammatory tissue reaction could not sufficiently be analyzed. It can be assumed that the HY addition has influence on the inflammatory alignment of the tissue reaction to a biomaterial to support the material-related healing processes, as has already been shown in other applications [32,33,34]. In this context, it is assumed that the application of a biomaterial that induces a more anti-inflammatory tissue reaction may also lead to improved bone tissue regeneration [35]. Macrophages and biomaterial-induced multinucleated giant cells (BMGCs) have been identified as key players in the tissue reaction to a biomaterial, as both cell types are involved in the biodegradation process as well as the inflammatory response within the implant bed [10,36]. Based on the above-described regenerative properties of HY, it is of special interest if the addition of this molecule to a BSP allows for the direction of the cellular basis of the tissue repair process. Interestingly, it has been revealed in more detail that HY fragments of different molecular sizes, i.e., high and low molecular weight hyaluronan (HMWHY/LMWHY), lead to different, sometimes opposing regenerative properties [37,38,39,40]. It has already been revealed that HMWHY exhibits anti-inflammatory and immunosuppressive characteristics, whereas LMWHY stimulates proinflammatory tissue reactions [21]. Thus, it can be expected that the combination of a BBSM with HMWHY might lead to a better consolidation of the bone healing process.

However, the question arises as to which amount of HMWHY is necessary to optimally stimulate the process of bone tissue regeneration. Thus, the aim of the present study was to analyze the influence of two different HMWHY amounts combined with a BBSM on the inflammatory tissue reaction in bone defects. For the in vivo study, the critical-size calvarian bone model in Wistar rats was used in combination with established and previously published histological, immunohistochemical, and histomorphometrical methods [41,42,43,44].

## 2. Results

### 2.1. Results of the In Vitro Cytocompatibility Analysis

The extract of the test material caused no toxilogical or biological damages to subconfluent monolayers of L929 cells under the test conditions of DIN EN:ISO 10993-5:2009. After 24 h incubation, there was no intracellular damage/alteration microscopically visible. The L929 fibroblasts showed discrete intracytoplasmatic granules, no cell lysis, and no cell growth inhibition. 

Moreover, the measurements of the cytotoxicity showed that in case of all extracts in all study groups, comparable values were found that did not significantly differ from the values of the negative control (Figure 1). Moreover, the values in the group of the positive control were significantly lower (*** *p* < 0.001) as assumed and expected (Figure 1).

Altogether, these results led to the conclusion that all of the tested biomaterials can be considered as non-cytotoxic and all of them meet the requirements of DIN EN ISO 10993-5.

### 2.2. Results of the In Vivo Study

#### 2.2.1. Results of the Histopathological Examinations

In all study groups, including the bone substitutes, the granules were found within their bony implantation beds without any microscopical signs of the added HMWHY in the respective study groups (Figure 2A–F). Moreover, the histological analysis showed that the granules of the BBSM were only partially integrated within newly-formed bone tissue in the peripheral regions of the implant beds neighboring the bone defect borders, without visible differences in the amounts of newly-formed bone in all study groups. Thus, the BBSM granules were embedded within a cell- and vessel-rich connective tissue, within the central regions of the critical-size defects (Figure 2A–F). In the control group also exhibited minor amounts of newly-formed bone, which were observable going out from the defect borders, while a thin layer of condensed connective tissue was found that seemed to cross the bone defect (Figure 2G,H). 

Additionally, the histopathological analysis revealed that most of the cells found in the implantation beds of the bone substitutes were macrophages, besides minor amounts of granulocytes, lymphocytes, and fibroblasts (Figure 2A–F). In the control group, mainly fibroblasts and macrophages have been observed (Figure 2G,H). At the surfaces of the BBS granules in the respective study groups, mainly macrophages, besides single biomaterial-induced multinucleated giant cells (BMGCs), were found (Figure 2A–F). No visible differences in the numbers of BMGCs have been observed in the study groups, including the BBS. In contrast, no multinucleated cells were found in the control group (Figure 2G,H).

Moreover, the histopathological analysis of the immunohistochemically-stained sections revealed that comparable high numbers of CD163-positive macrophages were found in the groups of the pure BBS, as well as in the group of the BBS combined with the low concentrations of HMWHY (Figure 2A,C). Lower numbers of CD163-positive macrophages were found in the group of the BBS combined with the high HMWHY concentrations, and in the control group without biomaterial insertion (Figure 2E,G). Moreover, comparable numbers of CD206-positive macrophages were detected within all study groups, including the control group (Figure 2B,D,F,H). Furthermore, the histopathological analysis showed that comparable numbers of both proinflammatory CD163- and anti-inflammatory CD206-positive BMGCs were found in all study groups containing the BBS (Figure 2A–F).

#### 2.2.2. Results of the Histomorphometrical Measurements

The histomorphometrical analysis showed that comparably high numbers of CD163-positive macrophages were found in the groups of the pure BBS (185.1 ± 54.43 cells/mm^2^), of the BBS combined with the low concentration of HY (173.0 ± 89.72 cells/mm^2^), and of the BBS combined with the high concentration of HY (112.6 ± 56.3 cells/mm^2^) (Figure 2). The highest numbers were detected in the groups of pure BBS and of the BBS combined with the low concentration of HY, which were both significantly higher (* *p* < 0.05 and ** *p* < 0.01) compared to the number of CD163-positive macrophages in the control group (54.15 ± 35.24 cells/mm^2^) (Figure 3). Thus, the only group whose values did not significantly differ in comparison to the number of CD163-positive macrophages in the control group was the study group of the BBS combined with the high concentration of HY (Figure 3). Moreover, comparably high numbers of CD206-positive macrophages were found in all study groups (Figure 3). Thus, in the group of the pure BBS a mean of 110.9 ± 26.06 cells/mm^2^ was found, in the group of the BBS combined with the low concentration of HY, 94.9 ± 53.2 cells/mm^2^, in the group of the BBS combined with the high concentration of HY, 98.53 ± 65.19 cells/mm^2^, and in the control group 115.3 ± 67.24 cells/mm^2^ were measured (Figure 3). 

The histomorphometrical analysis of the occurrence of biomaterial-induced multinucleated giant cells (BMGCs) showed that comparable total numbers of this cell type were found in the implantation beds of all three study groups (Figure 4). Thus, a mean of 11.31 ± 3.02 BMGC/mm^2^ was detected in the group of the pure BBS, while in the group of the BBS combined with the low concentration of HY a measurement of 9.76 ± 2.63 BMCGs/mm^2^ was detected and in the group of the BBS combined with the high concentration of HY 12.86 ± 65.19 BMGCs/mm^2^ were measured (Figure 4).

Moreover, comparable amounts of both CD163- and CD206-positive BMGCs were found in all study groups, without any significant differences between the numbers in the different groups nor the numbers of the different subtypes of the respective study groups (Figure 4). Thus, a mean of 2.81 ± 1.18 CD163-positive BMGCs/mm^2^ were found in the group of the pure BBS, while in the group of the BBS combined with the low concentration of HY it was 2.31 ± 1.47 CD163-positive BMCGs/mm^2^ and in the group of the BBS combined with the high concentration of HY 3.59 ± 2.06 CD163-positive BMGCs/mm^2^ were measured (Figure 4). Additionally, a mean of 5.18 ± 1.56 CD206-positive BMGCs/mm^2^ were detected in the group of the pure BBS, while in the group of the BBS combined with the low concentration of HY it was 5.85 ± 1.30 CD206-positive BMCGs/mm^2^ and in the group of the BBS combined with the high concentration of HY 6.47 ± 1.67 CD206-positive BMGCs/mm^2^ were detected (Figure 4).

## 3. Discussion

In the last decades, it has been revealed that the immune system is a very important factor for (bone) tissue healing [45,46]. In this context, it has been shown that correlations between the cytokine expression pattern of migrating macrophages and also so-called osteomacs, which are bone-resistent tissue macrophages, are of enormous importance for the material-mediated bone healing process [47]. In this context, it has been elucidated that nearly every biomaterial induces an inflammatory tissue reaction, including cells such as macrophages, as key regulators of the extent of inflammation that occurs as a response to the entirety of the physicochemical properties of a bone substitute materials (BSM), and especially of synthetic BSM based on calcium phosphates [10,48,49,50]. Moreover, it has been discussed that material-associated inflammatory granulation tissue also includes biomaterial-induced multinucleated giant cells (BMGCs) that are the fused-end stage of macrophages. These are involved in the biodegradation process of the BSM and are—like their mononuclear precursors—involved in the healing cascade by expressing both pro- and also anti-inflammatory molecules, such as the vascular endothelial growth factor (VEGF) or the heme oxygenase 1 (HO-1) [6]. Thus, the composition of a bone substitute is of enormous importance to guide the regeneration process and underlying processes that support bone tissue healing. However, there is little knowledge about the effect of BSM on the immune environment and its underlying mechanisms. 

Interestingly, it has already been shown that bone substitute pastes (BSP), which are mixtures of “classic” BSM in granular form, with hydrophilic and water-binding molecules such as hyaluronic acid (HY), support an integration behavior that is in accordance with the concept of guided bone regeneration (GBR) [4]. Moreover, the combination of BSM granules with such molecules helps to improve their applicability [2]. HY is of special interest as a component of biomaterials in general, and also for BSP, by providing additional regenerative features via modulation of processes such as wound healing, cellular migration, and angiogenesis via specific receptors such as the CD44 molecule [4,24,25,26,27,28,29]. It has already been shown in an in vivo study that a BSP containing β-TCP granules and a liquid phase combined of HY and methylcellulose triggers implant bed vascularization as an important factor of bone tissue regeneration [4]. In this context, high molecular weight hyaluronan (HMWHY) in particular is believed to trigger anti-inflammatory and immunosuppressive characteristics, in contrast to low molecular weight hyaluronan (LMWHY) that stimulates proinflammatory tissue reactions [21]. Thus, it can be expected that the combination of a BBSM with HMWHY might lead to better consolidation of the bone healing process.

However, in the aforementioned studies, the influence of the addition of HMWHY onto the material-associated inflammatory tissue reaction was not sufficiently analyzed. It can be assumed that the addition of HY has an influence on the inflammatory alignment of the tissue reaction to a biomaterial, to support the material-related healing processes as has already been shown in other applications [32,33,34]. In general, it is assumed that the application of a biomaterial that enables a stronger anti-inflammatory tissue reaction may also lead to improved bone tissue regeneration [35]. Thus, it is of special interest if the addition of this molecule to a BSP allows for the directing of the cellular basis of the tissue repair process. 

Therefore, the design of this study was to analyze the influence of two different amounts of HMWHY combined with a BBSM on the cytocompatibility and the inflammatory tissue reaction in bone defects. In particular, the identification of pro- and anti-inflammatory macrophages and BMGCs in the in vivo part of this study was conducted following established und previously published histological and histomorphometrical methods [30,41,42,44,51]. 

Initially, the in vitro analysis of the cytocompatibility showed that none of the materials caused toxilogical or biological damage to the L929 cells. Furthermore, the cytotoxicity measurements revealed that in case of all extracts, no differences between the values in the three study groups were found. This is not directly in accordance with the literature, as it has been shown in different studies that HY induces the proliferation as well as the collagen and protein synthesis of fibroblasts via interaction of the molecule, especially with the receptor for HY-mediated mobility (RHAMM) [52,53,54]. The results might on the one hand be explainable by the test procedure, which included extract preparation and not a direct contact of the cells with the materials. Thus, it is conceivable that the solubility of the HMWHY that was added to the BBSM granules in water was too low to have an influence on the cellular activity. However, another explanation is also possible: It is known that the HY molecule induces cellular responses to growth factors and cell migration, particularly for fibroblasts [55,56,57]. Thus, the added HMWHY might have more influence on the expression pattern and the mobility of the fibroblasts, rather than on the proliferation. However, these analyses were not part of the in vitro testing setup, and must be examined in a further study. Interestingly, these analysis steps are also not a part of the ISO norms, which demonstrates the limitations of these test procedures. Based on these facts, it might be reasonable to extend the related ISO norms in the future. 

The results of the in vivo study part showed that the BBSM granules were detectable within their implant beds in all respective study groups, while no histological signs of the added HMWHY were found. Interestingly, comparable amounts of newly-formed bone were detected in all study groups including bone substitutes, in contrast to the control group that contained considerably less regenerated bone matrix. Moreover, the histological analysis of all samples demonstrated that all three materials integrated within the implantation bed without inducing an undesired severe foreign body reaction. It was demonstrated that the implantation beds consisted of BBSM granules embedded in a cell- and vessel-rich connective or granulation tissue. Most of the cells found in the implantation beds of the bone substitutes were macrophages, and particularly at the BBSM granule surfaces there were mainly macrophages, besides the single biomaterial-induced multinucleated giant cells (BMGCs) that were detected. In contrast, a thin layer of condensed connective tissue containing mainly fibroblasts and macrophages has also been observed.

Additionally, both histopathological and histomorphometrical analyses of the immunohistochemically-stained sections revealed that comparably high numbers of CD163-positive macrophages were found in the groups of the pure BBS, as well as in the group of the BBS combined with the low concentrations of HMWHY. At the same time, lower numbers of CD163-positive macrophages were found in the group of BBS combined with the high HMWHY concentration and in the control group without biomaterial implantation. However, significant differences were only found between the cell numbers in the groups of pure BBS and of BBS combined with the low concentration of HMWHY, which were both significantly higher compared to the number of CD163-positive macrophages in the control group. Furthermore, comparable numbers of CD206-positive macrophages were detected within all study groups including the control group. Finally, the analyses showed that comparable total numbers and comparable numbers of both proinflammatory CD163- and anti-inflammatory CD206-positive BMGCs were found in all study groups containing BBS.

Altogether, these results lead to the conclusion that the higher concentration of HMWHY induced a minimal decrease of the pro-inflammatory tissue response, while no influence on the anti-inflammatory reactivity of the three materials was found. This means that all analyzed materials seem to have comparable levels of regenerative potential, which has also been reflected by comparable levels of newly-formed bone into their implantation beds. However, these results lead to the question of which effects the decrease of the pro-inflammatory response might have for the process of bone tissue regeneration.

On the one hand, it is concievable that this decrease might have an influence on the resorption of the BBSM granules via mononuclear phagocytes, such as the analyzed macrophages. In this context, it is known that both macrophages and BMGCS are involved in material degradation via phagocytosis [10,58]. Moreover, the process is (most often) associated with an expression of pro-inflammatory molecules, such as reactive oxygen species (ROS) and the tartrate-resistant acid phosphatase (TRAP), which are involved in the cellular degradation process of biomaterials, and especially of BSM [59]. Thus, a decrease in the pro-inflammatory tissue response might also reduce the resorbability of the materials. This might reduce the resorption pattern of a biomaterial so that a correlation with the concept of “creeping substitution” might no longer be possible [5,60]. Interestingly, a reduced resorption time might be also be favorable, even in case of patients with an impaired (cell) metabolism or healing time, such as tumor patients, diabetics, or elderly persons.

However, the unchanged induction of (pro-inflammatory) BMGCs seems to rebut this presumption, as this multinucleated cell type is especially induced by large foreign bodies for the purpose of their degradation via phagocytosis [36,41,61]. As no reduction of their pro-inflammatory subtype was detected, the resorption pattern of the analyzed BSP might not be influenced. However, little knowledge exists about the process of material degradation and especially about the phagocytosis capacity of macrophages and BMGCs. Further studies including more exact visualization methods, such as transmission electron microscopy (TEM), should investigate this phenomenon to get more insights into cellular interactions with biomaterials and especially with BSMs. 

Beyond this, it is conceivable that the decrease of the pro-inflammatory tissue response is an indicator for a preponderance of anti-inflammatory reactivity. This would mean that the addition of higher concentrations of HMWHY leads the tissue response towards an M2-driven reaction, which has assumed to be correlated with tissue healing [62]. Thus, a higher level of bone regeneration might have been induced after the time point analyzed in the present study. In this context, several studies examined how HY with different molecular weights affected the activation and reprogramming of macrophages [27,37,63,64]. These results suggest that macrophages undergo phenotypic changes dependent on the molecular weight of HY and support the conclusion that high molecular weight HY is mostly anti-inflammatory, whereas LMWHY induces a pro-inflammatory cell alignment and works as an alarm signal to the immune system. However, again, the unaltered response of both pro- and anti-inflammatory could be seen as an indicator of the unchanged alignment of the tissue response to the analyzed BSPs.

Interestingly, other studies showed similar results. For example, an in vivo study by Agrali et al. revealed similar findings to the present study. The authors showed that HY used alone or in combination with a resorbable collagen membrane and a bovine-derived xenograft did not contribute significantly to bone regeneration in rat calvarial bone defects [65]. Similar findings were presented by Maus et al., declaring that the application of an injectable hyaluronic acid (Hyalart^®^), with and without a bone morphogenetic protein 2 (BMP-2) addition, is not advantageous as a sole bone substitute for the filling of osseous defects [66]. 

An explanation for the similar cellular responses within the three groups could be the fast metabolization of the added HMWHY. Hence, the exogenous HY addition may be less stimulatory to cells, because of limited contact with the cellular environment and the absence of local biomechanical stimuli when compared to direct incorporation of HY into a biomaterial. Since HY is rich in carboxyl and hydroxyl groups, it can easily be modified through crosslinking or photo-crosslinking. HY should, therefore, be incorporated in one of the promising approaches based on a range of crosslinking techniques [21,67]. 

However, the present study also had limitations, which led to limited information about the degree of the inflammation response that was won by the counting of cells. In this context, the quantification of the expression of pro-and anti-inflammatory molecules might lead to further clarification of the tissue response induced by the HMWHY addition. However, more complex methods, such as laser-assisted cell microdissection, which allows for the measurement of cytokine release from single cells or cell types, are needed for examination of the cytokine levels released from biomaterial-induced macrophages and BMGCs. Moreover, the time span of up to 60 days might not be sufficient to evaluate the long time effects of the addition of HMWHY and its support in bone regeneration, so that further studies including more a prolonged observation period may also be needed. 

Altogether, the results of the present study showed that the tissue response in both the pure BBSM and the BSP are more or less comparable, with only slight differences in the pro-inflammatory response of macrophages, which might be induced be the addition of high concentrations of HMWHY. Moreover, the results showed that all analyzed materials are biocompatible, and the handling properties of the BBSM were improved by the molecule addition. 

## 4. Materials and Methods

### 4.1. Biomaterials

For preparation of the BBS, a mixture of granules of a synthetic biphasic calcium phosphate, i.e., 60% HA and 40% β-TCP, with a porosity of approx. 80%, a granule size of 0.5–1.0 mm, and 50 mg or 100 mg of hyaluronate with a molecular weight of 2 MDa was prepared (Table 1). The resulting bone grafting materials were sterilized by ethylene oxide (EO) and packed. The final biomaterials had a powder-like appearance and were hydrated before application to get a gel-like consistency.

### 4.2. In Vitro Cytocompatibility Analysis

The biphasic bone substitute and the negative control group (polypropylene) were immersed in DMEM culture medium (Thermo Fisher Scientific Inc., Waltham, Massachusetts, USA) during 24 h to prepare an extraction medium. After the extraction procedure, the extract was centrifuged at 2500× *g* for ten minutes to eliminate the residual sample. This was necessary to avoid a possible undesirable influence of the material extract onto the cells. 

A fibroblast (L929) suspension was prepared in complete DMEM supplemented with 10% fetal calf serum (FCS; Pan-Biotech GmbH, Aidenbach, Germany) to obtain 1,5 × 10^5^ cells/mL. The wells of 96-well tissue culture plates were inoculated with 100 μL of cell suspension per well. The plates were incubated for 24 h in a humidified incubator in 5% CO_2_ at 37 Co.

A 100 μL aliquot of the test material of different concentrations (100%, 66%, 44%, 30%, and 20%) were administered into each of the four wells following 24 h of incubation (5% CO_2_, 37 Co, 98% humidity). Afterwards, the cell culture plates were examined microscopically using a subjective grading system, assessing morphological aspects.

After microscopical examination, the culture medium was eliminated and 120 μL per well of MTS (3-(4,5-dimethylthiazol-2-yl)-5(3-carboxymethonyphenol)-2-(4-sulfophenyl)-2H-tetrazolium) was added. After 40 min, the colorimetric reaction in the MTS-treated plates was measured by spectrophotometry (absorbance at 490 and 630 nm). Cell viability was calculated from assay results divided by those of the negative control.

### 4.3. In Vivo Biocompatibility Analysis

#### 4.3.1. Experimental Animals, Surgical Procedure, and Explantation Procedure

In total, 20 female, 6–8 week-old Wistar rats obtained from the Military Medical Academy (Belgrade, Serbia) were randomly allocated into four study groups with 5 animals per group (*n* = 5) after authorization by the local Ethical Committee (Faculty of Medicine, University of Niš, Serbia). The in vivo experiments and animal housing were conducted at the Faculty of Medicine (University of Niš, Serbia). The animals were kept under standard conditions (water ad libitum, artificial light, and regular rat pellets) and standard pre- and postoperative care was ensured.

The animals obtained implantation of the aforementioned biomaterials for one study time period of 15 days. The implantations were conducted as follows. Initially, the animals were anesthetized via an intraperitoneal injection (10 mL ketamine (50 mg/mL) with 1.6 mL Xylazine (2%)) and the implantation side was shaved and disinfected. Afterwards, the surgical field was prepared by midline sagittal incisions combined with anterior and posterior subperiosteal dissection. Following this, the frontal and parietal regions of the calvarias were exposed and bilateral cranial bone defects (2 defects/calvaria) using a trephine bur (GC, Tokyo, Japan) were created. Afterwards, in the material groups, the biomaterials were inserted into both bone defects (Figure 5), while the no materials were implanted in the control group. Finally, the wounds were covered by a collagen membrane (Jason membrane, botiss biomaterials GmbH, Berlin, Germany) and sutured. 

After a healing period of two months, the explantation and the histological preparations were conducted as previously described [6,30,41,42]. Initially, the implanted bone substitutes were cut out, together with the peri-implant bone tissue, immediately after euthanasia and these explants were fixed in 10% neutral-buffered formalin for 48 h.

#### 4.3.2. Histological Preparations and Staining Methods

The histological preparations started with a decalcification step via 10% ethylenediaminetetraacetic acid (Fluka, Germany) at room temperature for 7–10 days. After that, the further histological work was performed by subsequent dehydration in a series of increasing alcohol concentrations, followed by xylol. All explants were cut in two halves and subsequently embedded in paraffin. Then, sections with a thickness of 3–5 µm were cut using a rotation microtome (Leica RM2255, Wetzlar, Germany). Next, immunohistochemical staining was performed as previously described by Barbeck et al. and Korzinskas et al. [43]. In brief, one section of every biopsy was stained via hematoxylin and eosin (HE), while four sections of every tissue explant were used for the immunohistochemical detection of the M1- and M2-subforms of phagocytes, i.e., macrophages and biomaterial-induced multinucleated giant cells (BMGCs). This was doen using antibodies against the pro- and anti-inflammatory molecules’ hemoglobin scavenger receptor (CD163) and mannose receptor (MR, also known as CD206). These slides were pretreated with citrate buffer and proteinase K at pH8 for 20 min in a water bath at 96 °C. Afterwards, equilibration using TBS-T buffer was conducted and the slides were then treated by H_2_O_2_ and avidin- and biotin-blocking solutions (Avidin/Biotin Blocking Kit, Vector Laboratories, Burlingame, CA, USA). Subsequently, incubation with the respective primary antibody for 30 min was done, followed by incubation with the secondary antibody (goat anti- rabbit IgG-B, sc-2040, 1:200, Santa Cruz Biotechnology, Shandon, CA, USA). After this step, the avidin–biotin–peroxidase complex (ThermoFisher Scientific, Dreeich, Germany) was applied for 30 min. Finally, counterstaining by hematoxylin and blueing was conducted.

#### 4.3.3. Histopathological Analysis

The histopathological analysis focused on the comparison of the tissue reactions to the different biomaterials on basis of the immunohistochemical stainings for the detection of the M1- and M2-subforms of phagocytes, i.e., macrophages and biomaterial-induced multinucleated giant cells (BMGCs). For this analysis, a light microscope Axio Scope.A1 (Carl Zeiss Microscopy GmbH, Germany) was used and an established protocol was applied as previously published [30,41,42,44,51]. In brief, the qualitative histological evaluation included the observation of the cells participating in the process of biomaterial integration and degradation, implantation bed vascularization and possible adverse reactions, such as fibrotic encapsulation or necrosis. A light microscope (Nikon ECLIPSE 80i microscope, Tokyo, Japan) was used for the analyses. Histological figures were taken by a microscope camera (Nikon DS-Fi1, Tokyo, Japan) that was connected to an acquisition unit (Nikon digital sight control unit, Tokyo, Japan).

#### 4.3.4. Histomorphometrical Analysis

The histomorphometrical analyses included comparative measurements of the numbers of the pro- and anti-inflammatory mono- and multinucleated phagocytes, on basis of previously published methods [30,41,42,44,51]. Briefly, so-called “total scans” were generated by means of a specialized scanning microscope, which is a combination of an Axio Scope.A1 (Carl Zeiss Microscopy GmbH, Germany) equipped with a digital camera and an automatic scanning table (Merzhaeuser, Germany) connected to an PC system running the Zen Core software (Zeiss, Tokyo, Japan). The resulting images were composed of 100 to 120 single images, with a 100× magnification in a resolution of 2500 × 1200 pixels and contained the complete implant area of the different bone substitutes, as well as the peri-implant tissue. To measure the numbers of cells within the bone defects, the complete area of the defect sides was first calculated with the “area tool” (in mm^2^). Afterwards, the different cells were manually marked using the “counting tool”. Finally, cell numbers per mm^2^ per defect area were determined by calculating the respective cell number in relation to the total implant area. 

### 4.4. Statistical Analysis

Quantitative data are shown as mean ± standard deviation after an analysis of variance (ANOVA) and a LSD post-hoc test using the GraphPad Prism 7.0d software (GraphPad Software Inc., La Jolla, CA, USA). Statistical differences were designated as significant if p-values were less than 0.05 (* *p* ≤ 0.05), and highly significant if P-values were less than 0.01 (** *p* ≤ 0.01) or less than 0.001 (*** *p* ≤ 0.001). 

## Figures and Tables

**Figure 1 ijms-20-01969-f001:**
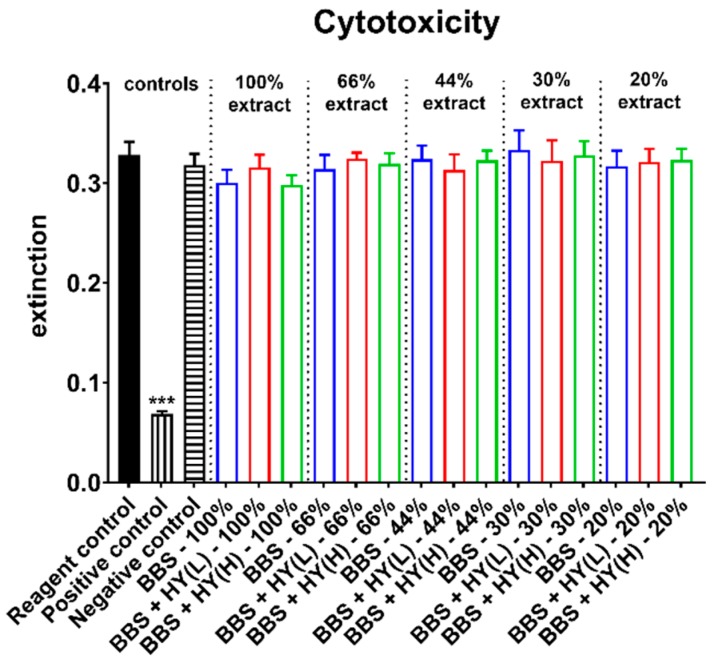
Results of the cell viability measurements of L929 fibroblasts used as measure for cytotoxicity (*** *p* < 0.001).

**Figure 2 ijms-20-01969-f002:**
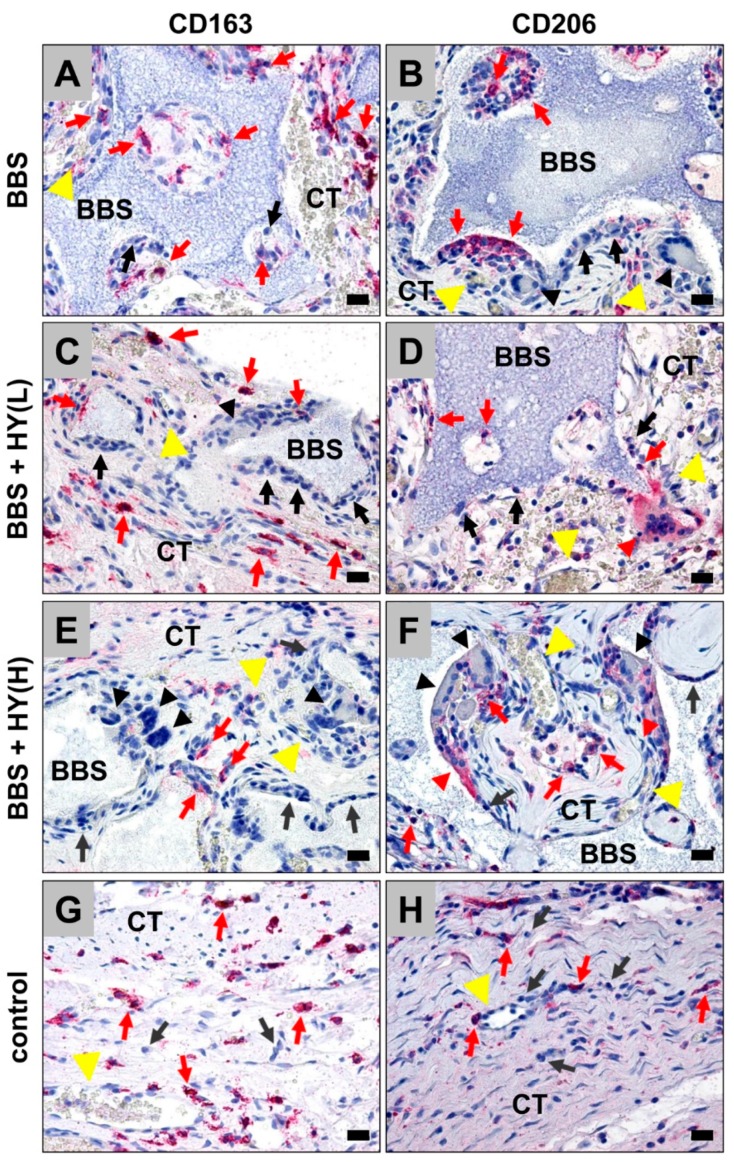
Exemplary histological images from the implantation beds of the analyzed pure biphasic bone substitute (BBS) (**A**,**B**) and the groups of the BBS combined with low (BBS + HY(L), (**C**,**D**)) and high (BBS + HY(H), (**E**,**F**)) concentrations of high molecular weight hyaluronan (HMWHY) embedded in connective tissue (CT) and the control group (**G**,**H**). CD163- and CD206-positive macrophages (red arrows), negative macrophages (black arrows), and CD163- and CD206-positive (red arrowheads) and -negative (black arrowheads) biomaterial-induced multinucleated giant cells (BMGCs) (vessels = yellow arrows) (CD163- and CD206- immunostainings, 200× magnifications, scale bars = 20 µm).

**Figure 3 ijms-20-01969-f003:**
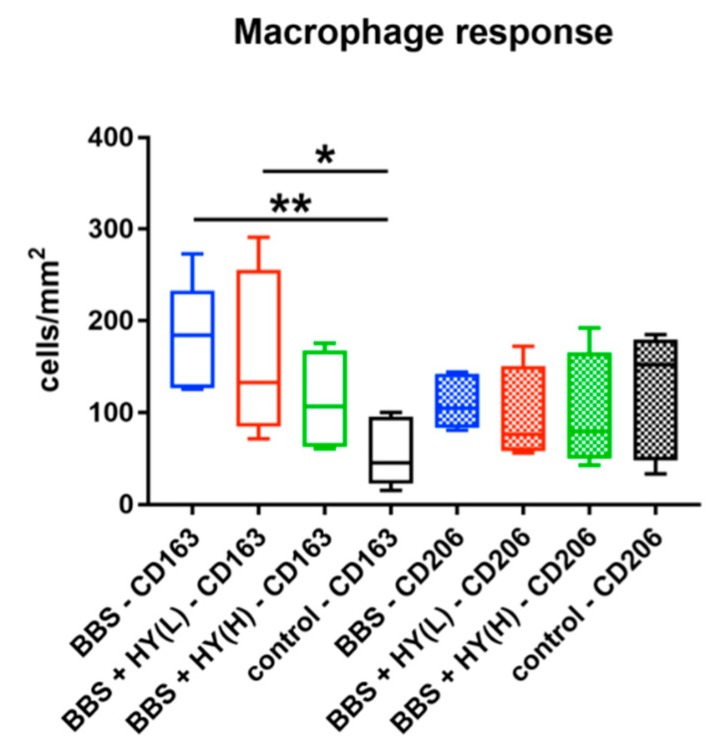
Results of the histomorphometrical analysis of CD163- and CD206-positive macrophages (* *p* < 0.05 and ** *p* < 0.01).

**Figure 4 ijms-20-01969-f004:**
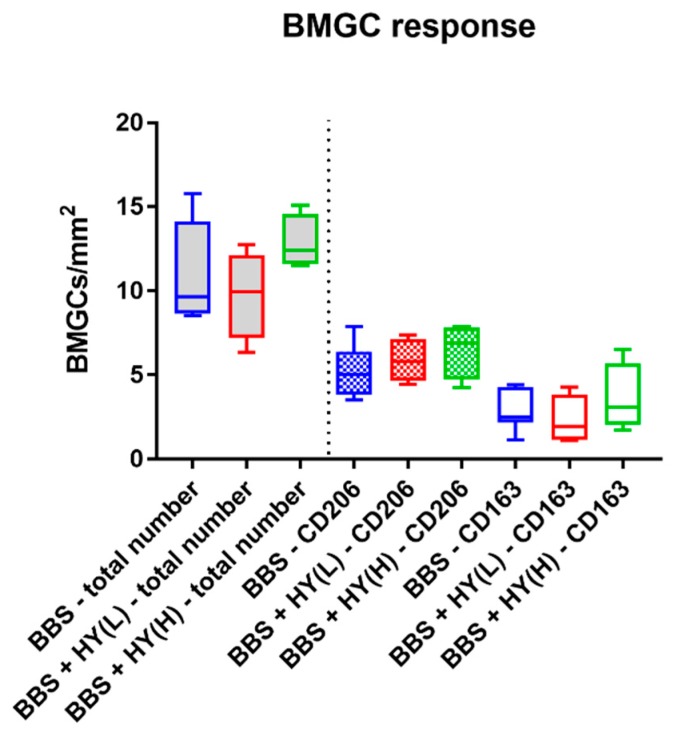
Results of the histomorphometrical analysis of the total number of biomaterial-induced multinucleated giant cells (BMCGs) (left side, grey fillings) and their CD163- and CD206-positive subtypes (right side).

**Figure 5 ijms-20-01969-f005:**
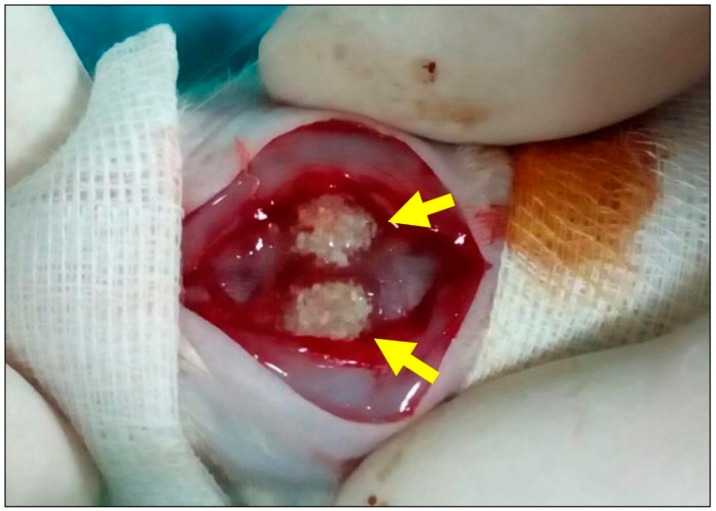
Sample image of the surgical procedure. Two calvarial defects (arrows) filled with biphasic bone substitute combined with the low HMWHY concentration.

**Table 1 ijms-20-01969-t001:** The different study groups and mixing ratios.

Study Groups	Group 1 (Control Operation)	Group 2 (BBS)	Group 3 (BBS + HY(L))	Group 4 (BBS + HY(H))
Biphasic bone substitute (BBS)	/	0,5 g	0,5 g	0,5 g
Hyaluronate, 2 MDa	/	/	50 mg	100 mg

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
