# Peer review of "The Addition of High Doses of Hyaluronic Acid to a Biphasic Bone Substitute Decreases the Proinflammatory Tissue Response"

_ijms, 2019, doi:10.3390/ijms20081969_

Reviewer 1 Report

The paper by Sieger et al. “The addition of high doses of hyaluronic acid to a biphasic bone substitute decreases the proinflammatory tissue response” studies the effect of adding high molecular weight hyaluronic acid to biphasic bone substitutes in terms of biocompatibility and inflammatory tissue reaction, concluding that high amounts of hyaluronic acid are not cytotoxic and lead to a decrease in pro-inflammatory macrophages, while also improving the material handling properties. The study is meaningful and the experiments are well carried out.

My main concern regards the “handling properties” that are not analyzed. The main conclusion states that adding HMWHY improves the material handling properties; however, no data is shown regarding the handling properties. How did HMWHY affect the handling properties? Does it affect the mechanical properties of the BBS? How was this measured or assessed?

Furthermore, in the results section, L140-144, the description of Fig2 should be more detailed, showing on the images what to look for: for instance, where are the vessels (“vessel-rich”)? Where are the “peripheral region” and the “bone defect borders”? Where is the “newly formed bone”? etc…

I also have a few minor comments, mostly regarding the English, listed below (I recommend the authors have a native English speaker check the quality of the English before resubmission):

L51-52: “to achieve their flow ability”: this isn’t very clear and should be reformulated

L75: “obtain”: “show” may be better.

L79: “have suspected“: “have been suspected” or “are suspected” is probably more correct

L87: “it has also shown”: “it has also been shown”

L95: “these both studies” should probably be “these two studies” or “both these studies”

L103: “have identified”: maybe “have been identified” is better

L107-108: “it has more detailed analyzed HY fragments”: this is not clear and should be reformulated

L122-123: “subconfluent monolayer”: “subconfluent monolayers” is more correct

L142: “that were neighbored to the bone defect borders” should probably be “neighboring the bone defect borders”

L142: “without visibly differences” should probably be “without visible differences”

L163: “comparable of both” should probably be “comparable numbers of both”

L201: “were found the group” should probably be “were found in the group”

L205: “were detected the group” should probably be “were detected in the group”

L225: “by expression both” should probably be “by expressing both”

L245: “was not sufficiently be analyzed” should be “was not sufficiently analyzed”

L260: “manifoldly”: do you mean “often” or “multiple times”?

L292: “in the in the” should be “in the”

L296: “that comparable of total numbers and of both” should be “that comparable total numbers and numbers of both”

L307: “are involved in to the material degradation” should be “are involved in the material degradation”

L312-314: “this might reduce…possible”: this sentence is difficult to understand and should be reformulated

L360: “will also ne” should be “will also be”

L370-371: why were 50mg and 100mg chosen for the “low” and “high” concentrations of HY?

L383: “The wells of 96 wells tissue culture plates”: “The wells of 96 well tissue culture plates” sounds better

L390: “After microscopically examination” should probably be “After microscopical examination”

Author Response

Dear reviewer,

many thanks for your uselful comments.

In the revised version we corrected the different mistakes.

Furthermore, we have no data regarding the handling properties of the IBS but most of the authors are clinicians in the field of maxillofacial surgery that use different bone substitues on a daily basis. Thus, the handling properties were evaluated on basis of their experiences and assessments. We hope this explanation is suitable for you!

As the manuscript is mainly focused on the inflammatory tissue response to the different materials we did not include images of the bony integration of the bone substitute granules at the defect borders as a variety of publications showed even images like this. Altogether, we only want to focus on the inflammatory cells and these images are included. However, we included arrows to indicate the vessels into the histological images now and hope that the manuscript is suitable for publication in the present version.

Many thanks and best wishes,

Mike Barbeck

Reviewer 2 Report

The manuscript by D. Sieger et al. describes the effect of hyaluronic acid on the biological response to biphasic bone substitutes. The data presented are useful for the design of of bone substitutes as well as provide new information about the host-biomaterials interactions. The manuscript is clearly written. 

In my opinion, the manuscript can be accepted for publication. The only minor suggestion is to better indicate what is shown in the Fig. 5. I recommend to point the zones of interest with arrows. 

Author Response

Dear reviewer many thanks for your useful comment! We revised the manuscript on basis of your comment and hope that it is suitable for publication int he present version.

Many thanks and best wishes,

Mike Barbeck